# Gels of Amyloid Fibers

**DOI:** 10.3390/biom9060210

**Published:** 2019-05-30

**Authors:** Ruizhi Wang, Xiaojing Yang, Lingwen Cui, Hang Yin, Shaohua Xu

**Affiliations:** Department of Biomedical & Chemical Engineering and Sciences, Florida Institute of Technology, Melbourne, FL 32901, USA; wangr2011@my.fit.edu (R.W.); xyang2008@my.fit.edu (X.Y.); lcui2011@my.fit.edu (L.C.); hyn2011@my.fit.edu (H.Y.)

**Keywords:** protein aggregation, amyloid plaque, Alzheimer’s disease, protein colloids, dementia, amyloidosis, atomic force microscopy

## Abstract

Protein self-assembly and formation of amyloid fibers is an early event of numerous human diseases. Continuous aggregation of amyloid fibers in vitro produces biogels, which led us to suspect that amyloid plaques and neurofibrillary tangles in Alzheimer’s disease are of biogels in nature. We applied atomic force microscopy, size exclusion chromatography, and differential scanning calorimetry to elucidate the gel’s structure, kinetics of gel formation, and melting point. We found that (1) lysozyme gelation occurs when the protein concentration is above 5 mg/mL; (2) nonfibrous protein concentration decreases and plateaus after three days of gel synthesis reaction; (3) colloidal lysozyme aggregates are detectable by both atomic force microscopy (AFM) and fast protein liquid chromatography (FPLC); (4) the gels are a three-dimensional (3D) network crosslinked by fibers coiling around each other; (5) the gels have a high melting point at around around 110 °C, which is weakly dependent on protein concentration; (6) the gels are conductive under an electric field, and (7) they form faster in the presence than in the absence of salt in the reaction buffer. The potential role of the gels formed by amyloid fibers in amyloidosis, particularly in Alzheimer’s disease was thoroughly discussed, as gels with increased viscosity, are known to restrict bulk flow and then circulation of ions and molecules.

## 1. Introduction

Protein self-assembly and formation of amyloid fibers and their deposition in tissues are thought to be responsible for numerous human diseases [1]. However, exactly how these fibers cause tissue damage and cell death, such as neurons in Alzheimer’s disease (AD), remains unknown [2]. For decades, the nonreactive amyloid fibers alone were assumed to be responsible for the pathogenesis. The fiber hypothesis remained unchallenged until the recent discovery of the loosely defined new structural feature oligomers or protein colloids. This discovery triggered a wave of effort over the last decade or so to better understand the oligomers, their cellular toxicity, and their potential as a therapeutic target for drug development against Alzheimer’s disease [3,4].

We report in this paper the study of gels of amyloid fibers (GAF), the formation and the characterization of the gel’s structure, thermal melting point, and conductivity to electric current. When amyloid fiber concentration in solution reaches a certain level, GAF is formed [5,6]. It remains unknown if and how GAF might affect cellular activity and tissue integrity. However, the potential of GAF to kill cells by restricting the movement of fluid, ions, nutrients, waste, and biosignaling molecules is significant mainly due to the lack of convectional flow or bulk flow through or within gels. GAF formation can increase local viscosity by as much as 10^10^-fold [7,8]. Gels can be impermeable to large molecules such as proteins, and this permeability is severely attenuated for small molecules such as water or ions [9]. A simple analogy is the restricted diffusion of large molecules such as proteins, DNA, or small molecules like bromophenol blue, in acrylamide or agarose gels in the absence of an electric field. If a protein or DNA is too large, it fails to enter the acrylamide gel. Since movement of molecules and ions is critical for cellular processes, such impact by GAF to tissue permeability can thus explain the cell death and tissue damage as seen in AD brain and systematic amyloidosis. Recently, we found that fibrin gels formed around sciatic nerves fibers attenuated the propagation of compound action potential [10]. It is unclear whether such attenuation is a result of restricted movement of ions, such as Na^+^ and K^+^, or molecules, such as oxygen, glucose and insulin.

Amyloid plaques or neurofibrillary tangles form gel-like structures [5,11,12]. Gels amyloid fibers are hydrogels by definition, a dispersion of molecules of liquid materials, such as water, within solid materials, such as amyloid fibers, in which the solid is the continuous phase and the liquid is the discontinuous phase [13]. If so, no fluid movement within the GAF is expected in a steady state. Hydrogels of many different materials are well characterized and applied broadly in society such as in food, breast implants, and drug delivery [14]. Nature hydrogels include mucus, vitreous humor, cartilage, tendons and blood clots. In vitreous humor, collagen fibers are thought to be evenly separated to a solid continuous phase and then to a gel due to electric charge repulsion [15].

Lysozyme is used to synthesize amyloid fiber gels, which allows us to search for methods that could be employed in the study of amyloid fiber gels in a fast and inexpensive way. These methods can then be applied to the study of amyloid fiber gels formed by other proteins or peptides, including Aβ. Protein fibers are known to form gels when the fiber concentration is high enough, such as gelatin gelation. From a biophysics perspective, hydrogels affect fluid flow. This allows us to propose that amyloid plaques of AD, potentially a hydrogel, might affect fluid movement and then circulation of molecules to and from neurons. Fluid flow plays an important role in the transport of ions and molecules for cellular homeostasis.

Lysozyme is used for the study of GAF, including the synthesis, property and structure analysis. It takes grams of proteins for the study of GAF performed in the paper, which would be too expensive if we use Aβ or tau. Since amyloid fiber formation is a generic property of proteins [16], lysozyme would be an ideal model protein for the study of GAF synthesis and characterization. Lysozyme is inexpensive and forms amyloid fibers in the human body, and its structure and properties are well characterized.

The characterization of GAF should be considered a new research direction for the amyloid community. The GAF can potentially be developed to simulate AD amyloid plaques for research and analysis. Currently, GAFs’ structure, properties, mechanism of formation, and role in the pathogenesis of various amyloid diseases are all unknown. Why can the extracellular matrix, which is also largely protein fibers, support cell growth while AD plaques cannot? What are the differences between gels formed by collagen fibers such as gelatin and amyloid fibers? How is GAF structurally different from the extracellular matrix? How are GAFs’ permeability to small and large biomolecules different from the extracellular matrix?

In this manuscript, we limit our scope of analysis to a few questions, as we are still at the beginning stages of GAF research for amyloidosis in general and AD in particular. We will focus on the analysis of (1) GAFs’ structures and synthesis; (2) the melting point; and (3) the conductivity to small ions. The answers to these questions might shed light on how the formation of amyloid fibers causes cell death in different tissues.

## 2. Materials and Methods

### 2.1. Materials 

Lysozyme from chicken egg white was purchased from Neova Technologies (Vancouver, WA, USA) and chemicals were purchased from Sigma-Aldrich Chemical Company (St. Louis, MO, USA).

### 2.2. Amyloid Fiber/Gel Synthesis

Unless specified, lysozymes of various concentrations were dissolved in water or 10 mM glycine, 150 mM NaCl solution, pH 2.5 [8,17]. For lysozymes dissolved in water, the charged amino acid residues in lysozyme provides pH buffer capacity and pH was adjusted using HCl. After the protein is completely dissolved, the samples were incubated in a shaker (Labnet, Vortemp 1550, Labnet International, Inc., Woodbridge, NJ, USA) at 500 rpm (revolutions per minute) at 55 °C. Samples were harvested at desired time points and frozen at −20 °C until analysis. Gel samples were removed by scooping the materials with a spatula.

### 2.3. Atomic Force Microscopy Imaging

To minimize atomic force microscopy (AFM) artifact from salt, fiber solutions made with lysozyme dissolved in water were used. The AFM samples were prepared as following a modified procedure of conventional specimen preparation for imaging [18,19,20]. A freshly peeled mica was first treated with 10 μL of 0.1 N NaOH and then samples of 10 μL of amyloid fiber solution or gel were applied to the mica surface. After a 10 min incubation under ambient condition, the mica was gently rinsed with 2 mL of distilled (DI) water by allowing water to flow slowly over the mica surface to remove salts and unbound proteins. Excess water and water droplets on the mica were removed by a gentle touch of Kimwipe tissue. The surface was first dried with a jet of dichloromethane applied tangentially or air dried overnight.

The AFM imaging was performed with a Molecular Imaging Picoscan Plus atomic force microscope (Molecular Imaging, Phoenix, AZ, USA) in contact mode using standard thin-leg silicon nitride cantilevers. Imaging generally began in a 5 × 5 nm^2^ area. When stable images were obtained, the scanning force was minimized by a reduction of the set point voltage, and the scanning area was increased to the desired size. Long and narrow legged standard Si_3_N_4_ cantilever tips were used. The raw data were plane-fit. Particle height was measured with the off-line data analysis program. Vertical and horizontal AMF measurements were calibrated with slides to which monodisperse NIST-traceable polyethylene spheres (Bangs Laboratories, Inc., Fishers, IN, USA) of 21, 30, 50, 200, and 450 nm diameter had been applied.

### 2.4. Fast Protein Liquid Chromatography (FPLC, Pharmacia) of Non-Fiber Lysozyme Species

An aliquot of 1.2 mL solution from lysozyme amyloid fiber reaction (20 mg/mL, pH 2.5) was transferred to a new Eppendorf tube at various time points, was centrifuged for 10 min at 12,000 rpm to remove fibers. Supernatant was transferred again to a new Eppendorf tube. 1 mL of the lysozyme supernatant was applied to the FPLC [21] with a Sephadex G-75 column (30 cm × 1.5 cm) equilibrated in 5 mM ammonium acetate buffer, pH 2.5. The protein was eluted with the same ammonium acetate buffer at a flow rate of 2 mL/min. Protein peaks were monitored using an OD280 monitor (Pharmacia, Stockholm, Sweden).

### 2.5. Melting Point Measurement

The melting point of the protein gel was determined using a differential scanning calorimetry (DSC, TA Instruments, New Castle, DE, USA). The DSC measures melting point by analyzing the relation between the heat absorbed per unit sample and the temperature increase for a sample in comparison with a reference.

Protein gel of known quantity, 10–20 mg, was placed in a metal pan for DSC. The pan was covered with a metal lid and sealed with a standard DSC Press from TA Instruments. An empty pan was used as a reference. Samples were scanned between 50 °C to 150 °C with a rate of 10 °C/min. A relation between the temperature (*x*-axis) and heat flow (*y*-axis, Walts/g) applied was presented. The corresponding temperature at the maximum heat flow is the melting point.

### 2.6. Conductivity Measurement

Lysozyme gels were prepared inside PTFE tubings (1/16’’ Inner diameter, 1/8” outer diameter, 55”, Grainger Co., Lake Forest, IL, USA). Lysozyme of different concentrations was prepared in glycine solution (10 mM glycine, 150 mM NaCl, pH 2.5). After filling with lysozyme solution, the tubings were sealed with a 3 cc syringe cap to prevent bubble formation and water evaporation and were incubated in a shaker oven (55°C, 500 rpm). A 10 cm tubing with GAF was used for conductivity measurement. The tubing was placed with each of its two ends in a test tube filled with buffer solution (10 mM glycine, 150 mM NaCl, pH 2.5). A voltage was applied, by inserting electrodes, to the two test tubes to generate an electric filed across the GAF tubing (EPS 300, Pharmacia Biotech, Piscataway, NJ, USA). Conductivities across the GAF tubing at various voltages were recorded by a digital multimeter (RadioShack, city, state (Melbourne, FL, USA).

## 3. Results

Amyloid plaques in vivo may well be in a gel state. Generation of GAF allows the simulation of amyloid plaques found in vivo and the analysis of the plaque’s property as a gel rather than individual fibers. GAF formed by lysozyme was analyzed for structure, relative concentration of fibrous and nonfibrous protein, conductivity under an electric field, melting point, and time dependence of fiber formation and gelation.

### 3.1. Gels of Amyloid Fibers Formation

We found that GAF can be formed under various conditions. Lysozyme concentration can be as low as 5 mg/mL, and no upper concentration limit was found if the protein can be dissolved in amyloidogenic buffer solution. High ionic strength of the solution promotes gel formation compared to conditions where the lysozyme was dissolved in water and pH adjusted with HCl. Both acidic pH and high temperature are essential for GAF synthesis, possibly through an elevated level of partial unfolding.

Prior to incubation, all samples were clear and soluble; no visible aggregates were present in any of the samples. After an extended period of incubation at 55 °C, different structures with distinct appearances had evolved in the samples (Figure 1, panel A). Fibrous aggregates formed in a very narrow range of pH 2.5 to 3.0. In the absence of salt, samples at pH 2.5 and lysozyme concentrations of 10 mg/mL and 20 mg/mL formed a mixture consisting of a clear, soluble phase and a phase of fibrous aggregates (Figure 1, Panel A, vials 1 and 2) after three weeks of incubation. When lysozyme (40 mg/mL) was incubated in the salt free solution, pH 2.5, 55 °C, the sample remained clear and gel free in the 1st week and turned into a gel-solution mixture in the 2nd week and a gel in the 3rd week. Samples at pH 2.5 and lysozyme concentrations of 70, 100 mg/mL transformed to a viscous, gelatin-like state referred to as a GAF (Figure 1, Panel A, vials 5 and 6). Such GAF structure is also formed at pH 3.0 at lysozyme concentrations of 40 and 70 mg/mL (Figure 1, Panel A, vials 3 and 4). The GAF has a white (vial 3) or light blue color (Figure 1A, vials 4 to 6) as compared to the transparent fiber solution (Figure 1A, vials 1 and 2). Addition of salt such as 10 mM glycine, 150 mM NaCl often leads to GAF formation within three days of incubation at pH 2.5, 55 °C.

### 3.2. Analysis of Gels of Amyloid Fibers 

The AFM analysis of the fiber solution revealed the presence of amyloid fibers as shown in Figure 1, panels B and C. Also apparent from the images is the presence of colloidal spheres to be present in amyloid fiber solution (Figure 1, panels B and C; and Figure 2, a in bottom panel). Although not a focus in this study, lysozyme fibers appear to be a linear aggregation of the colloids (Figure 1, panel B), which supports the linear colloidal aggregation model we introduced previously based on the study of human tau protein, yeast sup35, and mouse serum amyloid A [19,20,22,23,24].

The AFM images of fiber solution suggests coiling between fibers might be responsible for the high melting point observed for GAF. Occasionally, two fibers were found to self-assemble by merging to form a single larger fiber. Coiling may represent a mechanism for fiber-fiber interaction, in addition to the typical non-covalent intermolecular interactions, such as H-bonds, ionic interactions, and Van der Waals force. When lysozyme gels were diluted 1:100 in DI water and mixed gently by repeated tube inversion, the gel swelled considerably, and a small volume could be applied to mica in a monolayer (Figure 1 panel D). Most fibers appear to be uniform in diameter over distances of several micrometers. Few fibers, however, were found to have larger diameters than the majority, possibly due to the merging of small initial fibers of similar diameter to form larger fibers. The GAF on the other hand appears as a dense network of amyloid fibers (Figure 1, panel D). The spacing between fibers was 85.7 ± 5.8 nm (*n* = 60).

When the non-fibrous supernatant samples were applied to a size-exclusion chromatography FPLC (Sephadex G-75, Sigma-Aldrich), two peaks of different size were identified (Figure 2, middle panel). The first eluted peak was small and composed of mostly colloidal spheres and some short fibers. The second peak was large and composed mostly of monomeric lysozymes, dimers, and a small number of trimmers when analyzed by mass spectrometry. The AFM image of the second peak also reveals some colloids and short fibers. It is not clear if these aggregates were formed after the separation.

The amount of lysozyme in the sample that remained in non-fibrous form can be monitored over time. Such information can also be used to assess fiber formation rate and the amount of protein in fibrous or nonfibrous form. Amyloid fibers can be removed from a reaction solution by centrifugation (microfuge, 12000 rpm, 12 min). Centrifugal force is known to be enough to pellet fibers; however, small fibers with a length shorter than a couple hundred nanometers often remain in the supernatant together with protein colloidal spheres, oligomers and monomers. Protein concentration analysis of the supernatant by OD_280_ showed that non-fibrous protein concentration decreased slowly when fibers were formed and plateaued after three days of incubation (Figure 2, right Panel). It remains to be determined if these non-fiber forms of lysozyme are a result of unfinished reaction or the final equilibrium state of the reaction.

### 3.3. Gels of Amyloid Fibers’ High Melting Point

To study the nature of fiber-fiber interaction, DSC was performed on GAF. Upon heating, the gel proved to be thermoreversible as it gained a fluid consistency, suggesting that noncovalent bonds formed between fibers in GAF. The melting point was measured to be around 110 °C for GAF made with 10 mg/mL lysozyme in 10 mM glycine, 150 mM NaCl, pH 2.5 solution (Figure 3, Left Panel). The melting point is weakly dependent on lysozyme concentration, ranging between 109.5 °C and 114.5 °C for lysozyme concentration from 10 to 100 mg/mL, with a peak value around 114.5 °C for GAF made with 40 mg/mL lysozyme (Figure 3, Right Panel). The high melting point, compared to gelatin’s, suggests strong fiber-fiber interactions in GAF.

### 3.4. Gels of Amyloid Fibers’ Conductivity for Monovalent Ion Solution

Gels were made with lysozyme at different concentrations and conductivity of GAF was analyzed on different days of gelation. We found that the conductivity of GAF appears to be independent of fiber density and gelation time. Gels formed with lysozyme at concentrations between 5 to 100 mg/mL and gelation time of 3 to 30 days in 10 mM glycine, 150 mM NaCl, pH 2.5, were found to have nearly identical voltage and current relation (Figure 4). No significant differences were observed between tubes filled with GAF or buffers used to make the gels.

## 4. Discussion

The uncertainty of the pathogen that causes amyloid diseases in general and AD calls for a broad examination of molecules, molecular aggregates, and processes that are capable of attenuating cell function, action potential, plasticity, and survival. Much of the recent attention in amyloid research has been directed towards the toxicity of dimers and oligomers [25], intracellular a-beta accumulation [26], and the prion-like infectious nature of the amyloidogenic molecules [27]. We report here the identification, characterization, and mechanics of GAF as a potential pathogen in amyloid diseases. This report provides a new direction in searching for the cause of a group of human diseases where formation and deposition of amyloid fibers is an early event.

A number of original observations were made in this paper: (1) Individual fibers are capable of coiling around each other to cross link fibers; (2) GAF has a melting temperature of around 110 °C, which is higher than most gels made of biomaterials such as polysaccharides and gelatin; (3) GAF forms faster in the presence of 10 mM glycine and 150 mM NaCl than in water; (4) GAF is found to have a similar conductivity to simple ions such as Na^+^ and Cl^−^ as the buffer solution used to prepare gel; and (5) A significant amount of protein remains in non-fibrous form after gelation.

The protein concentration in the gels examined here is rather low compared to that in an amyloid plaque. Amyloid plaques from AD brain should have much higher fiber density than the GAF has. When isolated from AD brain tissue by ultracentrifugation, amyloid plaques migrate in a density around 1.2 to 1.4 g/mL [12,28]. A 10 mg/mL solution has a calculated density of 1.01 g/mL assuming the density of lysozyme to be 1.3 g/mL [29]. In other words, the protein quantity per unit volume is 20 to 40 times higher in a plaque than in a 10 mg/mL protein solution. Based on Figure 2, not all proteins form amyloid fibers. A substantial amount of proteins remains in the monomeric and colloidal form. Assuming only 50% of protein (10 mg/mL) in a reaction form fibers, the protein fiber density in a plaque would be 40 to 80 times higher in a plaque than in a gel. Assuming the core is three times as dense as the average of a plaque, the fiber density would be 120 to 240 times higher in the core of a plaque than in the gel formed from a 10 mg/mL protein solution. That means the actual pore size in the plaque core is 120 to 240 times smaller than the estimated pore size for the GAF. However, when the AFM image of the GAF was compared to the TEM image of neurofibrillary tangles, similar fiber separation was observed (Figure 5). Pore sizes in the two images are compatible. Does the structure integrity of the neurofibrillary tangles remain intact during their isolation?

Plaques in AD brains consist of dense fibers that stick together during isolation even in the presence of detergents [12,30]. Forces other than hydrophobic interaction must be involved among the fibers in the plaque. As shown in Figure 1, fibers can coil around each other to form connections through mechanical forces. As heat melts the gel, thermal energy is enough to disrupt most interactions between fibers. Noncovalent interactions including the mechanical force from coiling might play a primary role in fiber-fiber interaction. The coiling might explain the rather high melting temperature of GAF compared to gelatin, which melts at temperatures below 35 °C. Collagen in gelatin is known to have three helical fibers coiled around each other to form a coiled coil. We suspect individual fibers in GAF may crosslink more than two other fibers as found in collagen, and such crosslinking with many other fibers contributes to the high melting temperature.

We reported that earlier protein gels deposited by the axons attenuate the propagation of action potential [10]. The similar ion conductance across tubes filled with GAF or buffer solution shown in Figure 4 suggests that the ion movement is not attenuated by the presence of gels if an electric field is present. In vivo, the propagation of action potential and the oscillation of the membrane potential would provide such an electric field to drive ion movement. In theory, the presence of the gel around axons would restrict fluid flow and then circulation of ions to and from axons. Such interstitial fluid movement could be a result of cerebrospinal fluid or capillary fluid entering brain parenchyma as a result of hydrostatic and osmotic pressure. In addition to the ions or charged molecules, bulk flow would also contribute to the circulation of nonionic molecules such as oxygen, glucose, and macromolecules, which might be too large to diffuse through gels. Restricted circulation of the nonionic molecules by the gel may also contribute to the inhibitory effect of the gel on compound action potential propagation.

Amyloid plaque buildup occurs prior to neuronal death [32,33], which goes against the proposal that the neuronal toxicity was attributed to some intermediates or byproducts generated in fiber formation, including dimers, oligomers, and colloidal spheres. Should such intermediates or byproducts be toxic, neurons would be too unhealthy to produce a large amount of Aβ for plaque formation.

Amyloid plaques mostly damage neurites, axons, and dendrites traveling through the dense core, not the diffused fibers of the plaque [32,33]. Gel used for contact lenses is known to have a reduced O_2_ permeability [34]. Recall that neuronal damage occurs only when its neurites pass through the core of a dense plaque, not the diffused plaques [35]. One major difference between a dense core and diffused plaque is the limited transportation of molecules through the former (Figure 6). We recently reported that protein gels can attenuate the propagation of action potential [10] and introduced the denied access model that the neurite regions surrounded by amyloid gel plaques are denied from accessing to nutrient and signaling molecules, as well as to the drainage pathways for metabolic wastes. Our work suggests that an electric field can drive ions through the amyloid fiber gel. The neuronal membrane potential change might provide such an electric field and then a driving force for the movement of ions, but non-charged molecules such as glucose or O_2_ are insensitive to the electric field (Figure 6). It remains to be understood how such attenuated diffusion to neurites of O_2_ or other nutrient molecules affect neuronal function. Mitochondria in neurites provide ATP for the neurite function and metabolism; if O_2_ diffuses through neurite to mitochondria, then would the presence of the amyloid plaque around the neurite limit O_2_ availability and cause a significant drop in ATP production by neurite mitochondria?

Lysozyme is not a direct replacement of Aβ as the conditions for inducing their amyloid fiber formation is different and then the gelation conditions could be also different. However, the methods established here for studying lysozyme gels can be directly employed in the analysis of gels made with Aβ. It is known that Aβ forms amyloid fiber gels [5]. Also, all hydrogels share certain properties, especially their inhibitory effect on fluid flow, regardless of the nature of the fibers in the hydrogel, whether fibers of different kind of proteins or carbohydrate fibers. An example would be fibrin gels blocking blood flow, sealing bleeding, or causing stroke. Another example is fruit jello, a polysaccharide fiber capable of blocking fluid flow. Although mostly based on theoretical analysis, we like to use this paper to introduce our model that amyloid plaques are hydrogels which harm neurons by inhibiting the fluid flow and then the transport of molecules and ions to and from neurons.

If such a model protein is accepted, it will substantially accelerate our investigation of the pathogenesis of numerous human diseases where deposition of amyloid fibers is a common pathology hallmark. As re-crystalized lysozymes are easily available, this will save the research community an enormous amount of time in synthesizing and purifying the Aβ peptides or other proteins such as tau. It certainly would save us large sums of money. The critical question is whether lysozyme is a good model protein to study amyloid plaques or amyloid fibers formed in vivo by other proteins.

## Figures and Tables

**Figure 1 biomolecules-09-00210-f001:**
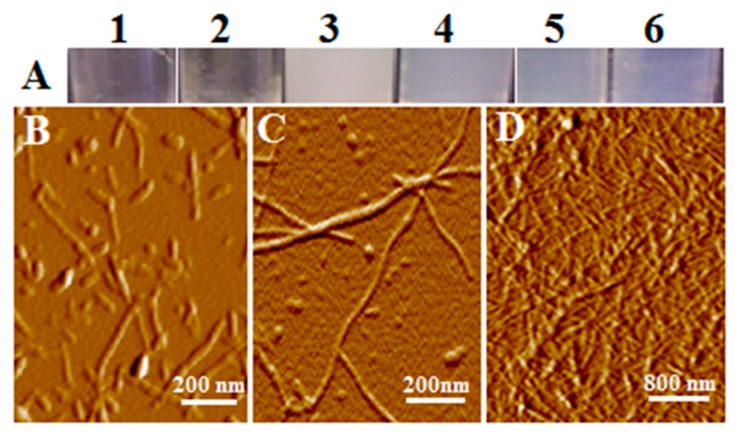
Atomic force microscopy (AFM) images of lysozyme fiber solution and gel. (**A**) visual appearance of the fiber solution (**1** and **2**) and gel (**3** to **6**). Gels appear in two colors, white (vial **3**) and light blue (vials **4**–**6**). Fiber solution and gels were formed by incubation of lysozyme under different concentrations at 55 °C, and pH 2.5, adjusted with HCl (vial **1**, **2**, **5**, **6**) and pH 3.0 (vial **3** and **4**). Vials (**1**) 10 mg/mL; (**2**) 20 mg/mL; (**3**) 40 mg/mL; (**4**) and (**5**) 70 mg/mL; (**6**) 100 mg/mL. (**B**–**D**) Representative AFM images are presented: (**B**) solution from vial **1**; (**C)** solution of gel diluted in water; (**D**) gel. Colloidal spheres, beaded chains, and the unbranching, linear morphology of the early stage fibers shown in panels **B** and **C** suggest a linear aggregation pathway. Panel **D** reveals a dense network of fiber structure piled on top of each other.

**Figure 2 biomolecules-09-00210-f002:**
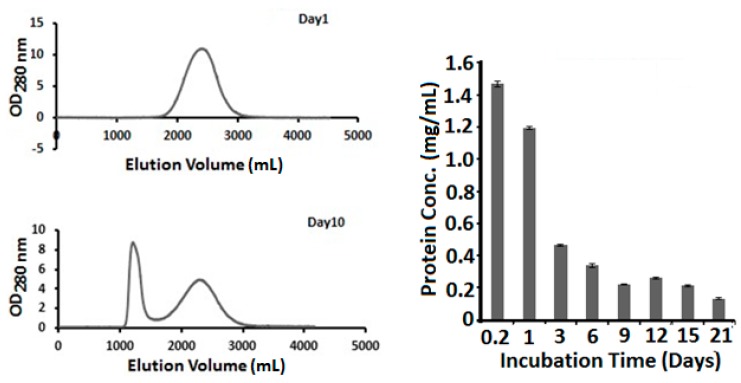
Non-fibrous lysozyme. As fibers and gel are formed, a substantial amount of lysozyme remains in non-fibrous form. (Left Panel) fast protein liquid chromatograpy (FPLC) analysis of soluble non-fibrous lysozyme. After centrifugation to remove fibers from the amyloid fiber solution, supernatant was analyzed by FPLC. The peak eluted at around 1200 mL from the column is thought to be the colloidal peak, consisting mostly spheres and very short fibers. The peak eluated at around 2300 mL is thought to be lysozyme monomers. (Right Panel) Non-fibrous lysozyme concentrations were measured for lysozyme fiber reactions harvested at different times.

**Figure 3 biomolecules-09-00210-f003:**
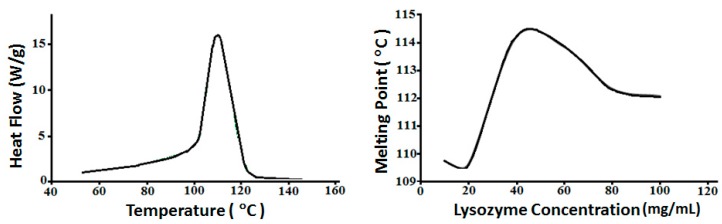
Differential scanning calorimetry (DSC) analysis of GAF melting point. Gels under this experiment were synthesised in 10 mM glycine, 150 mM NaCl, pH 2.5 solution. The melting point of lysozyme gel, 40 mg/mL, was analyzed using DSC. (Left Panel) Lysozyme gel shows a melting point around 110 °C. (Right Panel) Melting point varies for gels made with different protein concentrations with a range from 110 to 114.5 °C.

**Figure 4 biomolecules-09-00210-f004:**
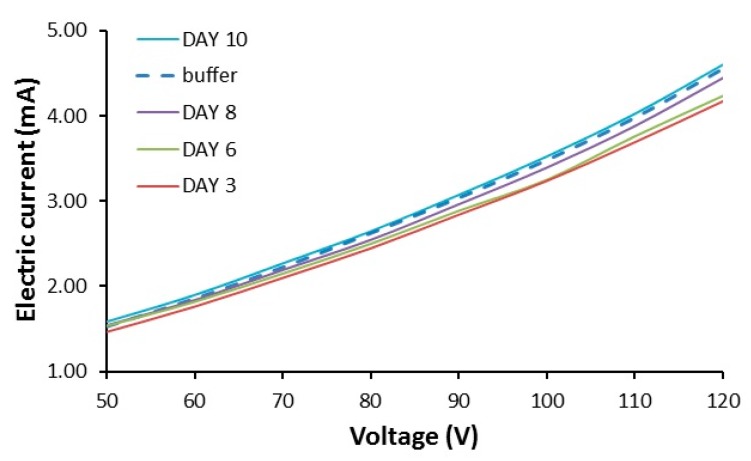
Conductivity of amyloid fiber formation. Five mg/mL, 10 mg/mL, 20 mg/mL, 40 mg/mL, 60 mg/mL, 80 mg/mL, 100 mg/mL lysozyme gels made with glycine solution (10 mM glycine, 150 mM NaCl, pH 2.5) were incubated in a shaker, 500 rpm, for 3 days. Currents through the tubes under different voltages were presented. Tubes filled with the glycine buffer were used as control.

**Figure 5 biomolecules-09-00210-f005:**
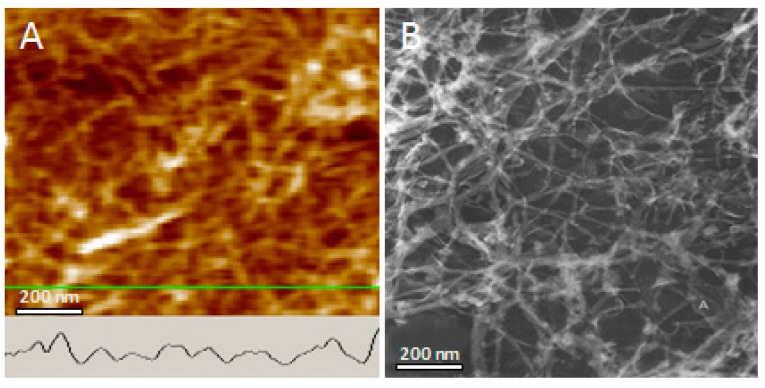
Network formed by lysozyme amyloid fibers (AFM, topographic view) and by tau neurofibrillary tangles (TEM image from Ruben, 1997). The AFM profile of lysozyme fiber is shown (inset). Distance between fibers and pore size is similar in both. Merging of fibers at points of contact suggest coiling of fibers with each other in both cases.

**Figure 6 biomolecules-09-00210-f006:**
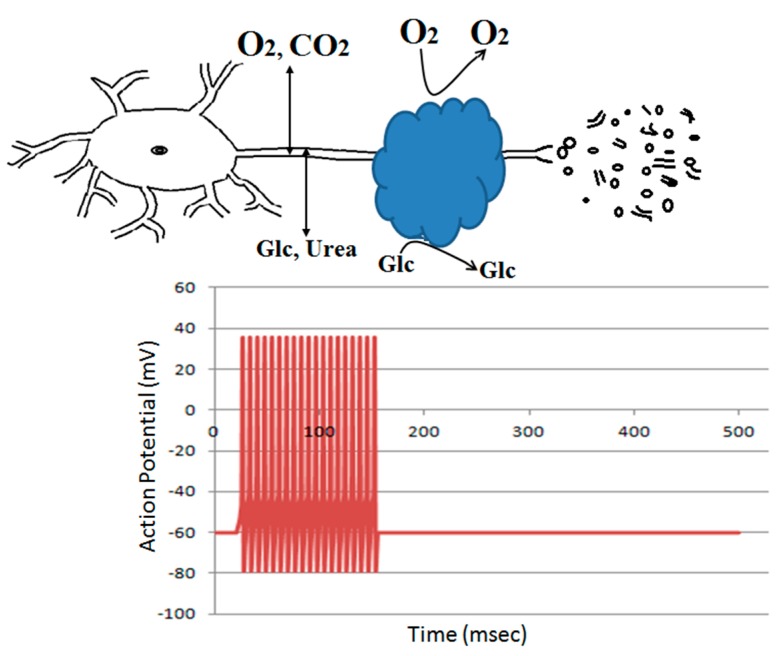
Denied Access Model. The neurite regions surrounded by amyloid gel plaques are denied from accessing to nutrient and signaling molecules as well as to the drainage pathways for metabolic wastes. This denied access leads to the inhibition of the action potential propagation and eventually the degeneration of the disused neurites [10,31].

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
