# Peer review of "Gels of Amyloid Fibers"

_biomolecules, 2019, doi:10.3390/biom9060210_

Round 1
Reviewer 1 Report
The manuscript required to include all the material or chemical used (sources) in the study and unable to provide the references for the Amyloid fiber/gel synthesis and FPLC.
Author Response
Responses to the 1st reviewer’s comments:
We thank the reviewer for the critical and constructive comments. We made revisions and hope the reviewer like the revised manuscript.
1: The manuscript required to include all the material or chemical used (sources) in the study and unable to provide the references for the Amyloid fiber/gel synthesis and FPLC.
Response: We took the reviewer’s suggestion and added the source where our lysozyme is purchased. We added two references for the synthesis of amyloid fiber/gel1,2. We also added one reference for FPLC methodology3.
(1) Woodard, D.; Bell, D.; Tipton, D.; Durrance, S.; Cole, L.; Li, B.; Xu, S.: Gel formation in protein amyloid aggregation: a physical mechanism for cytotoxicity. PloS one 2014, 9, e94789.
(2) Burnett, L. C., Burnett, B. J., Li, B., Durrance, T. D., and Xu, S. : A Lysozyme Concentration, pH, and Time-Dependent Isothermal Transformation Diagram Reveals Fibrous Amyloid and Non-Fibrous, Amorphous Aggregate Species. Open Journal of Biophysics 2014, DOI: 10.4236/ojbiphy.2014.42006.
(3) Madadlou, A.; O'Sullivan, S.; Sheehan, D.: Fast protein liquid chromatography. Methods in molecular biology (Clifton, N J ) 2011, 681, 439-47.
Reviewer 2 Report
The paper by Wang et al. uses a purely biophysical and simplistic approach to study synthetic lysozyme ‘biogels’, positing that these fibre-rich conglomerations of amyloidogenic protein fibres, criss-crossing each other, may model amyloid plaques or neurofibrillary tangles. The biogels were studied by AFM, FPLC, calorimetry, and conductivity experiments. The manuscript is highly speculative, follows on assumptions and hypotheses that have not led to any successful treatment for Alzheimer disease, and adds to ‘noise’ in an already-vast area of research which seems to have gone astray for years.
Major comment
Pathologically, dense amyloid plaques, diffuse amyloid plaques, or potentially inflammatory plaques {Shepherd, 2005 #14026} are not merely made of pure amyloid β-protein (Aβ); thus, a pure protein’s (lysozyme’s) biogel bound to a test tube would be far from modelling the complexity of the amyloid plaques or neurofibrillary tangles. This complexity does not refer only to composition, but also to mechanisms of formation and deposition of plaques and tangles. Besides, in the microenvironment of the brain parenchyma or cells, the components of plaques or tangles could expand or diffuse, or theoretically are unbound to do so, as they’re not bound within a highly concentrated vessel, such as a test tube. And their scales are a lot smaller than what can be produced in a test tube under artificial nonphysiological pH and shaking in short durations of time compared to the lengthy pathogenesis and progression of Alzheimer disease. The complexity of amyloid plaques (See a review published previously {Rahimi, 2018 #92441}) suggests that diverse deleterious mechanisms may underlie the pathogenesis or progression of Alzheimer disease—other than or additional to misfolding or deposition of Aβ, which has been modelled conventionally by speedy biophysical experiments without consideration of other disciplines that may explain, involve, or incriminate alternative mechanisms. It is still unclear whether protein misfolding is the cause or the outcome of disease progression {Morris, 2018 #92449}{Morris, 2014 #45800}. Thus, Aβ by itself is unlikely to be underlying AD pathogenesis or progression (see {Morris, 2014 #45800}{Morris, 2018 #92449}). Such controversies are validated by the failure or discontinuation of some high-profile clinical trials {Morris, 2014 #45800} designed based on the amyloid cascade hypothesis, forsaking the notion that Aβ is central to AD pathogenesis. It is likely that targeting Aβ (oligomers, fibres, or biogels) by some means may disrupt its potential physiological roles and may not be effective therapeutically in humans {Morris, 2014 #45800}. Thus, to this reviewer, the biogel preparation of lysozyme as a model of protein misfolding or in vivo deposition does not add anything new to an already failed hypothesis that has been seemingly perpetuated in the literature dogmatically representing Alzheimer disease research. I encourage the authors to frame and discuss their work as a purely biophysical exercise rather than extrapolating it as a model of plaque formation speculatively. A large set of arguments presented by Morris, Clark, and Vissel {Morris, 2018 #92449} may provide clues as to the new directions into researching Alzheimer disease rather than biogel formation.
Other comments:
1. Based on the major comment above, I encourage the authors to revise the sentence starting in line 7, which seems to be factually incorrect while considering all sides of the arguments about Alzheimer pathogenesis.
2. I would like the authors to discuss or speculate about what could cause a sudden highly concentrated accumulation of Aβ in the brain to cause biogel formation. Is a sudden accumulation possible for biogels to form in vivo? How could this be explained during the slow but progressive onset of the disease?
3. The authors suggest that formed biogels would interfere with the diffusion of oxygen, glucose, insulin or other nutrients to the cells. Does this mean that healthy (or otherwise) cells generate and secrete (in case of plaques) enough protein to cause biogel formation before delivery of nutrients can be disrupted and cells starve, lose function, and die? If this is the hypothesis, why does this happen in symptomatic patients and not in cognitively normal aged individuals who still harbour plaques? Why are cells driven to secreting high amounts of protein to cause biogel formation first before they get starved of nutrients, lose communication, and die? Is pathogenesis as simple as that? Or do cells die first before their contents are released to form biogels? In any case, if these are the postulated pathogenic mechanisms, one may ask why alternative mechanisms could not cause disease before plaque (biogel) formation? Why the diseased dead cells should contain so much amyloid protein that when they die, the biogel would form? Are these what the authors are suggesting for biogel as a model?
4. Revise lines 72 and 73 (257 and 258). The research into Alzheimer disease needs to be refocussed away from protein misfolding rather than focussed into another model of the same concept {Morris, 2018 #92449}. So much effort seems to have been wasted on the same concept based on amyloid cascade or oligomer cascade hypotheses {Hardy, 1992 #15065}{Hayden, 2013 #46072}, the latter without any major outcome to improve the quality of lives of the sufferers or their families’.
5. The authors should consider discussing how the large concentrations (molarity) of an amyloidogenic protein used to form biogels may relate to cell or plaque volume in vivo. How much protein should be produced/secreted in vivo to allow formation of a biogel the size of a plaque? And how long will this take considering the shaking and high pH may not be at work in vivo while other mechanisms to prevent such an accumulation may be at work? This model seems too artificial to be viable as a pathogenic mechanism. Considering all this and the progressive pathogenesis of the disease, how can fibre formation (or biogel formation) be an ‘early event’? Revise line 258.
6. How can research into biogel formation accelerate our understanding of Alzheimer disease while the many-decades-old amyloid cascade hypotheses have failed to lead to therapy? See line 343. It seems that the biogel study follows the same hypotheses but clothes them within a different, but gel-like, see-through cloak. I suggest the authors refocus their efforts to something more solid.
1. Shepherd, C.E.; Gregory, G.C.; Vickers, J.C.; Halliday, G.M. Novel 'inflammatory plaque' pathology in presenilin-1 Alzheimer's disease. Neuropathol. Appl. Neurobiol. 2005, 31, 503-511, 10.1111/j.1365-2990.2005.00667.x.
2. Rahimi, F. Aptamers Selected for Recognizing Amyloid β-Protein—A Case for Cautious Optimism. Int. J. Mol. Sci. 2018, 19, 10.3390/ijms19030668.
3. Morris, G.P.; Clark, I.A.; Vissel, B. Questions concerning the role of amyloid-beta in the definition, aetiology and diagnosis of Alzheimer's disease. Acta Neuropathol. 2018, 136, 663-689, 10.1007/s00401-018-1918-8.
4. Morris, G.P.; Clark, I.A.; Vissel, B. Inconsistencies and controversies surrounding the amyloid hypothesis of Alzheimer's disease. Acta. Neuropathol. Commun. 2014, 2, 135, 10.1186/s40478-014-0135-5.
5. Hardy, J.A.; Higgins, G.A. Alzheimer's disease: the amyloid cascade hypothesis. Science 1992, 256, 184-185.
6. Hayden, E.Y.; Teplow, D.B. Amyloid β-protein oligomers and Alzheimer's disease. Alzheimers Res. Ther. 2013, 5, 60, 10.1186/alzrt226.
Author Response
Responses to the 2nd reviewer’s comments:
We thank the reviewer for writing a long response and the many critical comments. Since a new model is introduced in the paper, it is understandable that the reviewer has so many questions and concerns. We hope that after reading our response, the reviewer will understand our manuscript better. It takes many papers to establish a new model to a research community.
Overall comments:
1: “The paper uses purely biophysical and simplistic approach to study synthetic lysozyme “biogels, positing that these fibre-rich conglomerations of amyloidgogenic protens fibres, criss-crossing each other, may model amyloid plaques or neurofibrillary tangles.”
Response: We agree with the reviewer that we applied a biophysical approach in our study. We found nothing wrong with using a biophysical approach for complicated puzzles. Positron emission tomography scan is a biophysical approach used for AD diagnosis and research.
In regard to the simplistic approach, we assume the reviewer worries using lysozyme as a model protein to study amyloid plaque and neurofibrillary tangles. Early understanding of the cross-beta structure and the fibrous morphology of amyloid fibers are from the analysis of antibody light chain (IgG) and serum amyloid A (SAA)4,5. Later we found the knowledge we learned about IgG and SAA amyloidosis is applicable to other amyloid fiber proteins including A-beta. As Chris Dobson of Cambridge6 claimed that amyloid fiber formation is a general property of protein molecules. We can extend that generic nature from amyloid fiber formation to amyloid fiber gel formation by different proteins under appropriate conditions. Then using a model protein like lysozyme to study the aggregation of amyloid fibers and formation of amyloid plaques or gels will shed light to the biogel formation by other proteins or peptides including A-beta and tau, which are expensive and time consuming to purify. After all lysozyme can form amyloid fibers in human body and cause disease as well.
2: “The manuscript is highly speculative, follows on assumptions and hypotheses that have not led to any successful treatment for Alzheimer’s disease, and adds to “noise” in an already-vast area of research which seems to have gone astray for years.”
Response: We assume the author is concerned about the amyloid cascade hypothesis. We believe the amyloid cascade hypothesis is too broad for our understanding of Alzheimer’s pathogenesis and failed to identify a specific pathogen. We need to know the pathogen and the pathogenesis that cause the massive synapse loss to develop a cure after the right target. A better understanding of the pathogenesis is required, which is the objective of our paper. The model we proposed that the amyloid fiber gels deny neurons’ access to ions and nutrients is new to the “already-vast area of research”. We are not aware of a model that is similar to ours in literature. Our model is based on the knowledge developed largely in physics and engineering on hydrogel’s effect on fluid flow and then mass transfer. Bioengineers treat atherosclerotic plaque and thrombotic clot as hydrogels. We should treat amyloid plaques as hydrogels as well. Hydrogel’s inhibitory effect on fluid flow is well established, and then amyloid plaque’s effect on the circulation of nutrients and ions to the neurites buried in the amyloid plaques should be examined. The reviewer can call those knowledge “noise” but it is a solid area of science dealing with Transport Phenomena.
Since we are still in search for the cause of the large synapse loss in AD, it is plausible to look for pathogens beyond the plaques and tangles. However, searching for other pathogens and analyzing amyloid plaques and their role in AD pathogenesis should be performed simultaneously. This is because (1) our understanding of the plaques and their interactions with neurites remain limited; for example, their effect on fluid flow and then circulation of nutrients and waste molecules to and from neurites; (2) Aggregation of molecules are responsible for numerous human diseases, including thrombotic clot, atherosclerotic plaque, kidney stones, gout, arthritis, and gallbladder stones to name a few. Treatment to these diseases often involve a removal of the aggregates and for stroke at least in a timely manner; and (3) amyloid plaque density and spread in different regions of the brain correlate with the stages of AD.
As to“The field have gone astray for years”, we think that physicians have been working in the field for over a century, but biochemists and biophysicists are only involved in this area of research recently. We hope that the latter will turn the muddy field around and provide a clean and clear scenery, as they possess a strong background in chemistry, physics, and engineering.
Major Comments:
3: “Pathologically, dense amyloid plaques, diffuse amyloid plaques, or potentially inflammatory plaques are not merely made of pure amyloid β-protein; thus a pure protein’s (lysozyme’s) biogel bound to a test tube would be far from modeling the complexity of the amyloid plaques or neurofibrillary tangles. The complexity does not refer only to composition, but also to mechanisms of formation and the deposition of plaques and tangles. Besides, in the microenvironment of the brain parenchyma or cells, the components of plaques or tangles could expand or diffuse, or theoretically are unbound to do so, as they’re not bound within a highly concentrated vessel, such as a test tube. And their scales are a lot smaller than what can be produced in a test tube under artificial nonphysiological pH and shaking in short durations of time compared to the lengthy pathogenesis and progression of Alzheimer disease. The complexity of amyloid plaques (See a review published previously, Rahimi, 2018) suggests that diverse deleterious mechanisms may underlie the pathogenesis or progression of Alzheimer disease—other than or additional to misfolding or deposition of Aβ, which has been modelled conventionally by speedy biophysical experiments without consideration of other disciplines that may explain, involve, or incriminate alternative mechanisms.”
Response: We understand that amyloid plaque has a complex composition and morphology, and then likely different interaction with the neurites. They remain poorly understood, both structurally and pathologically. Biochemists and biophysicists like to simplify things so that they can learn things with certainty, not “astray”, and then move the field forward firmly. Reductionist approach is a good approach when dealing with complex puzzles like AD pathogenesis. Biochemists purify proteins and analyze their structure and property in the test tubes or Eppendorf tubes all the time, and they could add other components to the system one by one to learn each of their effect. As long as we can learn the fundamental chemistry and physics behind a biologic process, we can avoid decade-long experiments nor under physiologic conditions. Protein crystals are hardly formed under physiologic condition and we can still resolve their structure. That is also how biochemists figured out all the metabolic pathways in human body.
“a pure protein’s (lysozyme’s) biogel bound to a test tube would be far from modeling the complexity of the amyloid plaques or neurofibrillary tangles.” Can lysozyme form amyloid fibers in human body? Yes. Does lysozyme amyloid fiber continue to aggregate and form a hydrogel? Yes. Is amyloid plaque a hydrogel? Yes. Does hydrogel of different chemicals share some common physical properties such as their inhibitory effect on fluid flow? Yes. If our knowledge of the complex amyloid plaque remains so poor, do we need continue to investigate these plaques for AD pathogenesis? Yes.
4: It is still unclear whether protein misfolding is the cause or the outcome of disease. Thus, Aβ by itself is unlikely to be underlying AD pathogenesis or progression (see Morris, 2014; Morris, 2018). Such controversies are validated by the failure or discontinuation of some high-profile clinical trials designed based on the amyloid cascade hypothesis, forsaking the notion that Aβ is central to AD pathogenesis. It is likely that targeting Aβ (oligomers, fibres, or biogels) by some means may disrupt its potential physiological roles and may not be effective therapeutically in humans. Thus, to this reviewer, the biogel preparation of lysozyme as a model of protein misfolding or in vivo deposition does not add anything new to an already failed hypothesis that has been seemingly perpetuated in the literature dogmatically representing Alzheimer disease research.
Response: “It is still unclear whether protein misfolding is the cause or the outcome of disease. Thus, Aβ by itself is unlikely to be underlying AD pathogenesis or progression.” This critic lacks logic. Not knowing if protein misfolding is the cause or outcome of the disease does not rule out the potential role of A-beta or the amyloid plaque in AD pathogenesis. This paper is intended to address whether and how A-beta and the amyloid plaques are involved in AD pathogenesis. See also our response to your concern #2 for our thought on amyloid plaque’s role in AD pathogenesis.
As to if anything new from our work, is it new to consider amyloid plaques as hydrogels? Is it new to introduce a pathogenesis model that the amyloid plaques deny the access of neurites to nutrients in AD? In medicine, we call amyloid plaques or neurofibrillary tangles, in biochemistry, biophysics or bioengineering, the amyloid plaques or neurofibrillary tangles would be regarded as hydrogels or biogels. This new recognition of the plaques as hydrogels is important as theories and experimental approaches developed over the last century for hydrogels can be applied in our study of amyloid plaques and neurofibrillary tangles. We should focus on how a biogel harm a cell first and worry about the kind of amyloid fibers latter.
Other Comments:
5: “Based on the major comment above, I encourage the authors to revise the sentence starting in line 7, which seems to be factually incorrect while considering all sides of the arguments about Alzheimer pathogenesis.”
Response: We hope the reviewer understand the paper better, especially about our model about the potential role of amyloid plaque in AD pathogenesis.
6: I would like the authors to discuss or speculate about what could cause a sudden highly concentrated accumulation of Aβ in the brain to cause biogel formation. Is a sudden accumulation possible for biogels to form in vivo? How could this be explained during the slow but progressive onset of the disease?
Response: The brain has decades to accumulate these plaques and tangles. The fact is that amyloid plaques and neurofibrillary tangles accumulate in AD brain. We don’t want “to speculate what causes a sudden highly concentrated accumulation of them”. That is not the focus of this paper.
7: The authors suggest that formed biogels would interfere with the diffusion of oxygen, glucose, insulin or other nutrients to the cells. Does this mean that healthy (or otherwise) cells generate and secrete (in case of plaques) enough protein to cause biogel formation before delivery of nutrients can be disrupted and cells starve, lose function, and die? If this is the hypothesis, why does this happen in symptomatic patients and not in cognitively normal aged individuals who still harbour plaques? Why are cells driven to secreting high amounts of protein to cause biogel formation first before they get starved of nutrients, lose communication, and die? Is pathogenesis as simple as that? Or do cells die first before their contents are released to form biogels? In any case, if these are the postulated pathogenic mechanisms, one may ask why alternative mechanisms could not cause disease before plaque (biogel) formation? Why the diseased dead cells should contain so much amyloid protein that when they die, the biogel would form? Are these what the authors are suggesting for biogel as a model?
Response: Harbouring plaques and clinic symptom are related, but there might be more dots to connect in between. We start to harbor atherosclerotic plaque in our artery wall after we pass age 20, but that does not mean we all have atherosclerotic heart attack in our youth. Again, the focus of this paper is not when and how A-beta are generated. We only focus on how such plaques can harm the neurites buried inside.
8: Revise lines 72 and 73 (257 and 258). The research into Alzheimer disease needs to be refocussed away from protein misfolding rather than focussed into another model of the same concept (Morris, 2018). So much effort seems to have been wasted on the same concept based on amyloid cascade or oligomer cascade hypotheses {Hardy, 1992; Hayden, 2013), the latter without any major outcome to improve the quality of lives of the sufferers or their families’.
Response: Again we hope the reviewer to reconsider his/her view after reading our arguments about the role of amyloid plaques and neurofibrillary tangles in AD pathogenesis.
9: The authors should consider discussing how the large concentrations (molarity) of an amyloidogenic protein used to form biogels may relate to cell or plaque volume in vivo. How much protein should be produced/secreted in vivo to allow formation of a biogel the size of a plaque? And how long will this take considering the shaking and high pH may not be at work in vivo while other mechanisms to prevent such an accumulation may be at work? This model seems too artificial to be viable as a pathogenic mechanism. Considering all this and the progressive pathogenesis of the disease, how can fibre formation (or biogel formation) be an ‘early event’? Revise line 258.
Response: Again, this is not the focus of the paper to address how much protein should be produced/secreted in vivo to allow the formation of an amyloid plaque, nor how long will this take. A brain has decades to produce and accumulate the proteins and the fibers. The truth is that plaques and tangles are deposited in AD brain cortex, and buried inside those plaques are a large number of axons and dendrites. As to an “early event”, plaques and tangles are known to form early before clinical symptoms appear. Aggregation of molecules are found to be an early event for numerous human diseases including atherosclerotic plaques.
10: How can research into biogel formation accelerate our understanding of Alzheimer disease while the many-decades-old amyloid cascade hypotheses have failed to lead to therapy? See line 343. It seems that the biogel study follows the same hypotheses but clothes them within a different, but gel-like, see-through cloak. I suggest the authors refocus their efforts to something more solid.
Response: Not calling the amyloid plaque a hydrogel detaches AD research from the knowledge and methods accumulated in engineering on transport phenomena in gels and other porous materials. Our judgement is limited by the set of knowledge we possess and then our angle of view of the problem. From the perspective of physical sciences and biochemistry and biophysics, applying theories and methods learned about hydrogels could benefit research of AD pathogenesis. Can we ignore biochemistry, biophysics or bioengineering in our research of medicine?
(1) Woodard, D.; Bell, D.; Tipton, D.; Durrance, S.; Cole, L.; Li, B.; Xu, S.: Gel formation in protein amyloid aggregation: a physical mechanism for cytotoxicity. PloS one 2014, 9, e94789.
(2) Burnett, L. C., Burnett, B. J., Li, B., Durrance, T. D., and Xu, S. : A Lysozyme Concentration, pH, and Time-Dependent Isothermal Transformation Diagram Reveals Fibrous Amyloid and Non-Fibrous, Amorphous Aggregate Species. Open Journal of Biophysics 2014, DOI: 10.4236/ojbiphy.2014.42006.
(3) Madadlou, A.; O'Sullivan, S.; Sheehan, D.: Fast protein liquid chromatography. Methods in molecular biology (Clifton, N J ) 2011, 681, 439-47.
(4) Cohen, A. S.; Calkins, E.: Electron microscopic observations on a fibrous component in amyloid of diverse origins. Nature 1959, 183, 1202-3.
(5) Bonar, L.; Cohen, A. S.; Skinner, M. M.: Characterization of the amyloid fibril as a cross-beta protein. Proc Soc Exp Biol Med 1969, 131, 1373-5.
(6) Stefani, M.; Dobson, C. M.: Protein aggregation and aggregate toxicity: new insights into protein folding, misfolding diseases and biological evolution. Journal of molecular medicine (Berlin, Germany) 2003, 81, 678-99.
(7) Xu, S.; Bevis, B.; Arnsdorf, M. F.: The assembly of amyloidogenic yeast sup35 as assessed by scanning (atomic) force microscopy: an analogy to linear colloidal aggregation? Biophysical journal 2001, 81, 446-54.
(8) Xu, S.: Cross-beta-sheet structure in amyloid fiber formation. The journal of physical chemistry. B 2009, 113, 12447-55.
(9) Xu, S.: Aggregation drives "misfolding" in protein amyloid fiber formation. Amyloid 2007, 14, 119-31.
(10) Dauberman, W., Breit, S., Xu, S. : Protein Gelation around Axons Inhibits Action Potential Propagation in Nerve Fibers. Journal of Alzheimer's Disease & Parkingsonism 2017, 7, 1000349.
(11) Dauberman, W., Xu, S.: Alzheimer's disease pathogenesis: the denied access model. Journal of Alzheimer's Disease & Parkingsonism 2017, 7, 1000359.
(12) Benilova, I.; Karran, E.; De Strooper, B.: The toxic Abeta oligomer and Alzheimer's disease: an emperor in need of clothes. Nature neuroscience 2012, 15, 349-57.
(13) De Felice, F. G.; Wu, D.; Lambert, M. P.; Fernandez, S. J.; Velasco, P. T.; Lacor, P. N.; Bigio, E. H.; Jerecic, J.; Acton, P. J.; Shughrue, P. J.; Chen-Dodson, E.; Kinney, G. G.; Klein, W. L.: Alzheimer's disease-type neuronal tau hyperphosphorylation induced by A beta oligomers. Neurobiology of aging 2008, 29, 1334-47.
(14) Chromy, B. A.; Nowak, R. J.; Lambert, M. P.; Viola, K. L.; Chang, L.; Velasco, P. T.; Jones, B. W.; Fernandez, S. J.; Lacor, P. N.; Horowitz, P.; Finch, C. E.; Krafft, G. A.; Klein, W. L.: Self-assembly of Abeta(1-42) into globular neurotoxins. Biochemistry 2003, 42, 12749-60.
(15) Hardy, J.: The amyloid hypothesis for Alzheimer's disease: a critical reappraisal. Journal of neurochemistry 2009, 110, 1129-34.
Reviewer 3 Report
The authors Wang et al. present the kinetics and biophysical characterization of gels formed by the fibrils of lysozyme and propose a model of hydrogel formation by amyloid proteins like Aβ. They also speculate that the GAF, especially, GAF of Aβ might be responsible for the neuronal death by blocking fluid flow to the neurons. Thought interesting, there are some concerns that need to be addressed before the article can be considered for publication –
1. The authors acknowledge that the focus of the field has shifted from fibrils to oligomers and their toxic and prion-like nature. The authors also show that majority of the protein was in oligomeric/short fibrillar form even after 21 days of incubation indicating that oligomers and/or short fibrils form a major part of GAF. So, if authors propose that GAF are the pathogens responsible for neuronal death, what part of GAF is toxic; is it the oligomers/short fibrils or the dense matrix of fibrils or the ensemble of both? The reviewer suggests that authors present some form of toxicity studies to support their claim that GAF are, indeed, pathogenic. Several reports have been published that show for certain amyloid proteins like tau oligomers, not fibrils, are not toxic (refer Ghag et al. (2018) Protein Science, doi - https://doi.org/10.1002/pro.3499).
2. Protein gelation could be result of increasing concentration and not necessarily intrinsic to the amyloid nature of certain proteins. The authors mention several examples of other protein undergoing hydrogel formation. Did authors test, in their experimental set-up, the characteristics of gel formation by non-amyloidogenic proteins? Whether gel formation is a concentration-dependent only phenomenon or intrinsic to amyloid nature of proteins is an important question to consider and answer?
3. The authors should consider determining the tensile strength of the GAFs.
Other than these, following few minor points should be corrected –
1. In line 67, correct ‘expansive’ to ‘expensive’
2. In line 68, correct ‘A-beta’ to ‘Aβ’
3. Rephrase line 166-167 as GAF had two colors, white (Fig. 1A vial 3) & light blue (Fig 1A vials 4-6)
The article can be considered for publication with above mentioned minor revisions.
Author Response
Responses to the 3rd reviewer’s comments:
We thank the reviewer for the critical and constructive comments. We hope the reviewer like the revised manuscript.
1: “The authors acknowledge that the focus of the field has shifted from fibrils to oligomers and their toxic and prion-like nature. The authors also show that majority of the protein was in oligomeric/short fibrillar form even after 21 days of incubation indicating that oligomers and/or short fibrils form a major part of GAF. So, if authors propose that GAF are the pathogens responsible for neuronal death, what part of GAF is toxic; is it the oligomers/short fibrils or the dense matrix of fibrils or the ensemble of both? The reviewer suggests that authors present some form of toxicity studies to support their claim that GAF are, indeed, pathogenic. Several reports have been published that show for certain amyloid proteins like tau oligomers, not fibrils, are not toxic (refer Ghag et al. (2018) Protein Science, doi - https://doi.org/10.1002/pro.3499).”
Response: Although we found the spheres early in 1996 using AFM, and reported our discovery in 20011, we call them colloids in our previous papers2-4 as dozens of molecules are in each of the sphere. We detailed our argument why we are concerned about the A-beta oligomers’ (AβO) cell toxicity model5. The main issue with the oligomer hypothesis is the diversity of toxicity. Countless articles are published which present new mechanistic hypotheses of AßO toxicity. AßOs lack a common description of structural toxicity and are thought of as “an emperor in need of clothes” since they possess numerous conformations ranging from monomers, to trimers, and to eventual fibrils6 . On top of that, AβO are soluble and can be carried around by the body fluid and then we would have difficulty to explain their localized cell toxicity. Moreover, not all oligomers promote toxicity 7,8 and “relevant Aß toxicity has barely been demonstrated9. If AßOs are the toxic component, the massive cell loss, role in cognitive alterations, and existence in non-AD patients cannot be fully explained 6.
For GAF, we believe it’s the physical, not chemical, property of the hydrogel that eliminates bulk flow, causes dysfunctional neurites and we provided evidence in a previous paper demonstrating how the presence of a hydrogel could inhibit the propagation of action potential4.
2. “The authors should consider determining the tensile strength of the GAFs.”
Response: Yes, we agree with the reviewer. Many properties of GAF need to be characterized including the tensile strength. We will work on it in the future.
4: Other than these, following few minor points should be corrected –
1. In line 68, correct ‘A-beta’ to ‘Aβ’
3. Rephrase line 166-167 as GAF had two colors, white (Fig. 1A vial 3) & light blue (Fig 1A vials 4-6)
The article can be considered for publication with above mentioned minor revisions.
Response: We fixed these problems.
(1) Xu, S.; Bevis, B.; Arnsdorf, M. F.: The assembly of amyloidogenic yeast sup35 as assessed by scanning (atomic) force microscopy: an analogy to linear colloidal aggregation? Biophysical journal 2001, 81, 446-54.
(2) Xu, S.: Cross-beta-sheet structure in amyloid fiber formation. The journal of physical chemistry. B 2009, 113, 12447-55.
(3) Xu, S.: Aggregation drives "misfolding" in protein amyloid fiber formation. Amyloid 2007, 14, 119-31.
(4) Dauberman, W., Breit, S., Xu, S. : Protein Gelation around Axons Inhibits Action Potential Propagation in Nerve Fibers. Journal of Alzheimer's Disease & Parkingsonism 2017, 7, 1000349.
(5) Dauberman, W., Xu, S.: Alzheimer's disease pathogenesis: the denied access model. Journal of Alzheimer's Disease & Parkingsonism 2017, 7, 1000359.
(6) Benilova, I.; Karran, E.; De Strooper, B.: The toxic Abeta oligomer and Alzheimer's disease: an emperor in need of clothes. Nature neuroscience 2012, 15, 349-57.
(7) De Felice, F. G.; Wu, D.; Lambert, M. P.; Fernandez, S. J.; Velasco, P. T.; Lacor, P. N.; Bigio, E. H.; Jerecic, J.; Acton, P. J.; Shughrue, P. J.; Chen-Dodson, E.; Kinney, G. G.; Klein, W. L.: Alzheimer's disease-type neuronal tau hyperphosphorylation induced by A beta oligomers. Neurobiology of aging 2008, 29, 1334-47.
(8) Chromy, B. A.; Nowak, R. J.; Lambert, M. P.; Viola, K. L.; Chang, L.; Velasco, P. T.; Jones, B. W.; Fernandez, S. J.; Lacor, P. N.; Horowitz, P.; Finch, C. E.; Krafft, G. A.; Klein, W. L.: Self-assembly of Abeta(1-42) into globular neurotoxins. Biochemistry 2003, 42, 12749-60.
(9) Hardy, J. A.; Higgins, G. A.: Alzheimer's disease: the amyloid cascade hypothesis. Science 1992, 256, 184-5.
Round 2
Reviewer 2 Report
Thank you to the authors for responding to my comments. Good luck for future research.